# Intra-Tumour Heterogeneity Is One of the Main Sources of Inter-Observer Variation in Scoring Stromal Tumour Infiltrating Lymphocytes in Triple Negative Breast Cancer

**DOI:** 10.3390/cancers13174410

**Published:** 2021-08-31

**Authors:** Darren Kilmartin, Mark O’Loughlin, Xavier Andreu, Zsuzsanna Bagó-Horváth, Simonetta Bianchi, Ewa Chmielik, Gábor Cserni, Paulo Figueiredo, Giuseppe Floris, Maria Pia Foschini, Anikó Kovács, Päivi Heikkilä, Janina Kulka, Anne-Vibeke Laenkholm, Inta Liepniece-Karele, Caterina Marchiò, Elena Provenzano, Peter Regitnig, Angelika Reiner, Aleš Ryška, Anna Sapino, Elisabeth Specht Stovgaard, Cecily Quinn, Vasiliki Zolota, Mark Webber, Davood Roshan, Sharon A. Glynn, Grace Callagy

**Affiliations:** 1Discipline of Pathology, Lambe Institute for Translational Research, School of Medicine, National University of Ireland Galway, H91 TK33 Galway, Ireland; darren.kilmartin@nuigalway.ie (D.K.); M.OLOUGHLIN10@nuigalway.ie (M.O.); mark.webber@nuigalway.ie (M.W.); sharon.glynn@nuigalway.ie (S.A.G.); 2UDIAT-Centre Diagnòstic, Pathology Department, Institut Universitari Parc Taulí-UAB, Parc Taulí, 1, 08205 Sabadell, Spain; andreu.xavier@gmail.com; 3Department of Pathology, Medical University of Vienna, Währinger Gürtel 18-20, 1090 Vienna, Austria; zsuzsanna.horvath@meduniwien.ac.at; 4Division of Pathological Anatomy, Department of Health Sciences, University of Florence, 50134 Florence, Italy; simonetta.bianchi@unifi.it; 5Tumor Pathology Department, Maria Sklodowska-Curie National Research Institute of Oncology, Gliwice Branch, 44-102 Gliwice, Poland; Ewa.Chmielik@io.gliwice.pl; 6Department of Pathology, Bács-Kiskun County Teaching Hospital, 6000 Kecskemét, Hungary; csernig@kmk.hu; 7Laboratório de Anatomia Patológica, Instituto Politécnico de Coimbra, 3000-075 Coimbra, Portugal; pbsf@ipocoimbra.min-saude.pt; 8Laboratory of Translational Cell and Tissue Research, Department of Imaging and Pathology, University Hospitals Leuven, 3000 Leuven, Belgium; giuseppe.floris@uzleuven.be; 9Unit of Anatomic Pathology, Department of Biomedical and Neuromotor Sciences, University of Bologna, Bellaria Hospital, 40139 Bologna, Italy; mariapia.foschini@ausl.bologna.it; 10Department of Clinical Pathology, Sahlgrenska University Hospital, 41345 Gothenburg, Sweden; aniko.kovacs@vgregion.se; 11Department of Pathology, Helsinki University Central Hospital, 00029 Helsinki, Finland; paivi.heikkila@hus.fi; 122nd Department of Pathology, Semmelweis University Budapest, Üllői út 93, 1091 Budapest, Hungary; janinakulka@gmail.com; 13Department of Surgical Pathology, Zealand University Hospital, 4000 Roskilde, Denmark; anlae@regionsjaelland.dk; 14Department of Pathology, Riga Stradins University, LV-1007 Riga, Latvia; intaliepniecekarele@inbox.lv; 15Unit of Pathology, Candiolo Cancer Institute FPO-IRCCS, 10060 Candiolo, Italy; caterina.marchio@unito.it (C.M.); anna.sapino@ircc.it (A.S.); 16Department of Medical Sciences, University of Turin, 10126 Turin, Italy; 17Department of Histopathology, Cambridge University Hospitals National Health Service (NHS) Foundation Trust, Cambridge CB2 0QQ, UK; elena.provenzano@addenbrookes.nhs.uk; 18National Institute for Health Research Cambridge Biomedical Research Centre, Cambridge CB2 0QQ, UK; 19Diagnostic and Research Institute of Pathology, Medical University of Graz, 8010 Graz, Austria; peter.regitnig@medunigraz.at; 20Department of Pathology, Klinikum Donaustadt, 1090 Vienna, Austria; angelika.reiner@icloud.com; 21The Fingerland Department of Pathology, Charles University Medical Faculty and University Hospital, 50003 Hradec Kralove, Czech Republic; ryskaale@fnhk.cz; 22Pathology Department, Herlev University Hospital, DK-2730 Herlev, Denmark; elisabethidaspecht@gmail.com; 23Irish National Breast Screening Programme, BreastCheck, St. Vincent’s University Hospital, D04 T6F4 Dublin, Ireland; cquinn@svhg.ie; 24School of Medicine, University College Dublin, D04 V1W8 Dublin, Ireland; 25Department of Pathology, School of Medicine, University of Patras, 26504 Rion, Greece; zol@med.upatras.gr; 26School of Mathematics, Statistics and Applied Mathematics, National University of Ireland Galway, H91 TK33 Galway, Ireland; davood.roshansangachin@nuigalway.ie

**Keywords:** TILs, sTILs, triple negative, breast cancer, reproducibility, international immuno-oncology biomarker working group

## Abstract

**Simple Summary:**

The stromal tumour infiltrating lymphocytes (sTILs) within a tumour are a strong predictor of outcome for patients with triple negative breast cancer (TNBC). However, the assessment of sTILs is subject to variation and needs to be standardized in order for it to be used more widely as a biomarker. The aim of this study was to determine the level of consistency that can be achieved when an internet-based scoring aid is used to assist with evaluation of sTILs. Twenty-three breast pathologists across Europe scored sTILs in 49 cases of TNBC taken from a routine diagnostic practice using this aid. The consistency of scoring sTILs was good. However, variation in the distribution of sTILs within the tumour resulted in discordance between pathologists scoring cases, particularly as it caused variability in the selection of regions of the tumour to score. More rigorous training of pathologists is needed for standardization of sTILs assessment, which may potentially be improved using automated approaches.

**Abstract:**

Stromal tumour infiltrating lymphocytes (sTILs) are a strong prognostic marker in triple negative breast cancer (TNBC). Consistency scoring sTILs is good and was excellent when an internet-based scoring aid developed by the TIL-WG was used to score cases in a reproducibility study. This study aimed to evaluate the reproducibility of sTILs assessment using this scoring aid in cases from routine practice and to explore the potential of the tool to overcome variability in scoring. Twenty-three breast pathologists scored sTILs in digitized slides of 49 TNBC biopsies using the scoring aid. Subsequently, fields of view (FOV) from each case were selected by one pathologist and scored by the group using the tool. Inter-observer agreement was good for absolute sTILs (ICC 0.634, 95% CI 0.539–0.735, *p* < 0.001) but was poor to fair using binary cutpoints. sTILs heterogeneity was the main contributor to disagreement. When pathologists scored the same FOV from each case, inter-observer agreement was excellent for absolute sTILs (ICC 0.798, 95% CI 0.727–0.864, *p* < 0.001) and good for the 20% (ICC 0.657, 95% CI 0.561–0.756, *p* < 0.001) and 40% (ICC 0.644, 95% CI 0.546–0.745, *p* < 0.001) cutpoints. However, there was a wide range of scores for many cases. Reproducibility scoring sTILs is good when the scoring aid is used. Heterogeneity is the main contributor to variance and will need to be overcome for analytic validity to be achieved.

## 1. Introduction

Stromal tumour infiltrating lymphocytes (sTILs) have emerged as a strong prognostic factor in HER2 positive and triple negative breast cancer (TNBC) [1]. Data from trials and retrospective analyses have demonstrated that an increasing level of sTILs is associated with improved disease free and overall survival (OS) for patients with TNBC treated with adjuvant anthracycline-based chemotherapy [2,3,4,5,6] and improved response to neoadjuvant chemotherapy for TNBC and HER2 positive disease [7,8,9]. Recent preliminary data from the KEYNOTE-086 randomised phase II trial also suggest that sTILs may predict responses to pembrolizumab in metastatic TNBC [10]. On the basis of accumulating data, the 16th St Gallen International Breast Cancer Conference has recently attributed level 1B evidence to sTILs in breast cancer (BC) and has endorsed routine reporting of sTILs in TNBC [11].

sTILs are appealing as a biomarker because assessment can be performed on routine H&E slides by light microscopy at the time of diagnosis and requires no additional testing. Guidance for assessing sTILs was produced by the international TIL Working Group (WG), which is now known as the International Immuno-Oncology Biomarker-WG [12], and this guidance is recommended by the WHO as the methodology for quantifying sTILs in BC [13]. It advises that sTILs are assessed in treatment naive tumours, within the tumour boundary and as a continuous variable. The TIL-WG has also developed a comprehensive web-based resource (available online: www.tilsinbreastcancer.org (accessed on 25 February 2021) around TILs in cancer that includes training and learning tools to aid the practicing pathologist and to improve standardisation. The website includes an interactive scoring aid and a prognostic tool for TNBC into which the absolute sTILs value is incorporated in current practice, sTILs assessment is used for stratifying cases in clinical trials and in studies evaluating prognosis, and while assessment of sTILs is included in the pathology report in some centres, it is not yet included in minimum datasets for BC reporting.

For sTILs to be used more widely as a biomarker in routine clinical practice, the scoring methodology must be reproducible and meet the required standards for analytic validity. Whilst several studies have reported at least good levels of inter-observer agreement for visual assessment in BC [14,15,16,17,18,19], issues remain. One of the main challenges is the heterogeneous nature of sTILs, which is difficult to quantify. The extent to which this impedes standardising visual assessment and whether more sophisticated computational and machine learning approaches are required are uncertain. However, the TIL-WG has demonstrated impressive reproducibility assessing sTILs using their interactive internet-based scoring aid, which was superior to that achieved without it [14,18,19]. The key elements of this scoring aid, namely the requirement to evaluate multiple tumour areas and match them against calibrated sTILs reference images, are postulated to improve consistency by mitigating the effect of heterogeneity. The tool was designed for use as an educational and training resource, but given the excellent reproducibility reported when it was used in the TIL-WG reproducibility study and the challenges associated with consistency in practice [18], its potential to improve accuracy of scoring sTILs in clinical practice merits exploration.

The aim of the present study was to evaluate the consistency of sTILs assessment using the internet-based scoring aid in cases taken from routine practice. Although the tool was not designed for use in a daily practice setting, a secondary aim of this study was to give an indication of the potential for the tool, or elements of it, to be used by pathologists to score sTILs in routine practice. To this end, experienced breast pathologists from the European Working Group for Breast Screening Pathology (EWGBSP) used the scoring aid to evaluate sTILs in core biopsies taken from a diagnostic service in one institution. The Group previously undertook a similar study without the scoring aid [15], thereby facilitating a comparison of the level of agreement achievable with and without it. As in our previous study, cases were limited to TNBC because the potential clinical role for sTILs is strongest in TNBC and to eliminate differences between tumour subtypes as a contributor to any variation in scoring. 

## 2. Materials and Methods

### 2.1. Case Material

The material used for this study consisted of needle core biopsies from 49 consecutive cases of TNBC diagnosed between 2016 and 2017 taken from a single tertiary breast cancer centre (Appendix A). The criteria for inclusion were the presence of sufficient diagnostic material in the core biopsy. The majority of cases were invasive breast carcinoma NST and three were other types; most were grade 3. Triple negative status was confirmed by immunohistochemistry for oestrogen receptor (SP1), progesterone receptor (16/SAN27) and HER2 (4b5) and, where appropriate, by fluorescent in situ hybridisation for HER2. A median of 2 cores was present per case (range 1–4). A representative haematoxylin and eosin (H&E) stained 3 μm section from each case was selected. Whole slide images were captured by an Olympus VS120 digital slide scanner at a magnification of 40× and saved in their image file format. VSI. The digitised H&E-stained slides were anonymised and uploaded to the PathXL online platform.

### 2.2. Study Circulations

The study involved two circulations; twenty-three pathologists (median 24 years breast pathology experience, range 4–40) participated in the first and 14 of the original group undertook the second circulation. In the first circulation, participants were sent an email including a link to the PathXL repository of the 49 digitized slides. Participants were asked to score cases according to published guidance [12] using the tool accessible online at: www.tilsinbreastcancer.org (accessed on 1 February 2020). The guidance recommends that only sTILs within the tumour border are evaluated and scored in a semi-quantitative manner and that the final score represents the average sTILs across the slide not focusing on hot spots. The email sent to participants included a guide that explained how to use the tool including how to view slides, capture images, score sTILs, and report the results. Participants were required to select at least three separate tumour areas or fields of view (FOV) from the digital slide of the full biopsy but were free to include additional FOV at their discretion. The captured FOV were uploaded onto the tool, and each was scored by matching it against reference images at 200×–400× magnification, representing each 10% increment of sTILs. The tool averages the scores for the FOV from each case to give a final score for the case. 

Participants were asked to record sTILs scores in a Microsoft (MS) Excel template and to include comments regarding features deemed relevant to scoring the case i.e., heterogeneity, necrosis, issues with defining the tumour boundary, tumour cellularity, fragmentation, or any other relevant comment. Participants were asked to record the time taken to capture and to score each slide and to complete all cases and for a general comment on the usability of the tool. 

In the second circulation, a single set of three FOV from the same 49 cases (*n* = 147) was selected by one pathologist (GC) from the organising institution for all the other participants to score. The purpose of this exercise was to ascertain the relative contribution of the selection of tumour areas to score by the pathologist and scoring the FOV to variation. The FOV in circulation 2 were selected to represent the average sTILs of the case according to TILs-WG guidance [12] i.e., only areas within the boundary were included and zones of necrosis, regressive hyalinisation, normal lobules, and DCIS were avoided. FOV were not selected to represent low, intermediate, or high sTILs. For circulation 2, the FOV from the 49 cases were relabelled and randomly re-ordered such that they could not be linked back to the cases scored by the participants in the first circulation. The captured images of the FOV were then uploaded onto the web-based tool for each participant to score using the tool. This was completed over six months after the first circulation was complete. 

The scanning of slides, pseudo-anonymization, relabelling and randomisation of cases and collation of scores was performed by a technical officer (MW) who did not participate in the study. Pathologists did not have access to the scores of other pathologists in the study. 

### 2.3. Statistical Analysis

The pathologists’ sTILs scores were collated in MS Excel, with statistical analysis performed in Excel 2007 and IBM SPSS Statistics 25. The scores from each pathologist were compared to each other and across the two circulations, assessed as raw scores and as binary categories using a range of cutpoints. Box-and-whisker plots were used to show both the distribution of sTILs scores per pathologist and the distribution of sTILs scores per slide. The two-way mixed single measures intraclass correlation coefficient (ICC) was used to determine how similar the sTILs percentages scored by different pathologists were for each slide in the study [20,21]. The ICC used in this study was the two-way mixed-effects, consistency, single measure because the aim of the analysis was to examine the consistency of scores across different raters while considering a degree of systematic error (e.g., if they were generally high or low scores), rather than assessing how close the absolute scores were to one another, as a minor difference in absolute scores was considered to be less important [22]. An ICC value between 0.75 and 1.00 was considered as excellent inter-observer agreement; between 0.60 and 0.74 as good agreement; between 0.40 and 0.59 as fair agreement; and less than 0.40 as poor agreement [20]. The Spearman’s correlation coefficient (ρ) was used to measure how closely the raw sTILs scores given by each individual pathologist for the first circulation correlated with the raw scores given by the same individual pathologist for the second circulation. Colour-coded “heat maps” were generated for both circulations, visually portraying the array of individual pathologist sTILs scores for all cases. Statistical significance was determined by a result having a two tailed *p* value of less than 0.05.

## 3. Results

Twenty-three pathologists completed the first circulation and 14 of the 23 pathologists completed the second circulation. The average time taken to select FOV and upload images of the FOV was 5.5 min (median, 5, range 2–10) for each case. This was in addition to an average of 4.8 min to score the case (median 3.5 min, range 1–12 min).

### 3.1. Inter-Observer Agreement

For the first circulation, inter-observer agreement between the 23 pathologists for the absolute sTILs score for the 49 cases was good (ICC 0.634, 95% CI 0.539–0.735, *p* < 0.001). However, inter-observer agreement was only poor or fair when scores were represented in binary categories using a range of cutpoints from 10% to 60% with the best agreement observed using the 25% (ICC 0.573, 95% CI 0.475–0.684, *p* < 0.001) (Table 1).

In order to assess whether the inter-observer agreement between the 14 participants who completed the second circulation was representative of that of the original 23 in circulation 1, a separate analysis was performed using the scores for the cases from these 14 participants in circulation 1. Inter-observer agreement between the 14 pathologists for absolute sTILs score was good in circulation 1 (ICC 0.662, 95% CI 0.566–0.760, *p* < 0.001) and similar to the level of agreement observed for the original group of 23 pathologists. The level of agreement between these 14 pathologists for the binary categories in circulation 1 was also similar to that observed for the original 23 pathologists and again was slightly better for the 25% cutpoint (ICC 0.602, 95% CI 0.501–0.711, *p* < 0.001) than for the others (Table 1).

When a single pathologist selected representative FOV from each case and the images of these FOV were then scored by the other participants, the inter-observer agreement between 14 pathologists improved for absolute sTILs score and was excellent (ICC 0.798, 95% CI 0.727–0.864, *p* < 0.001). Agreement also improved for binary scores and was best for the 20% (ICC 0.657, 95% CI 0.561–0.756, *p* < 0.001) and 40% (ICC 0.644, 95% CI 0.546–0.745, *p* < 0.001) cutpoints. 

In order to obtain more granular data on inter-observer agreement, the same analysis was performed at FOV-level using the 147 FOV from circulation 2. Inter-observer agreement scoring individual FOV was almost identical to that measured at case level (*n* = 49) for both absolute sTIL scores (ICC 0.734, 95% CI 0.685–0.782, *p* < 0.001) and the binary categories (Appendix A). Thus, there was little change in agreement, as measured by ICC, between cases and FOV.

### 3.2. Distribution of Scores between Pathologists

The median sTILs scores for all pathologists were similar in circulation 1 (23 participants: 17%; 14 participants: 18%) and in circulation 2 (18%). However, there was variation in the distribution of scores between the individual pathologists (Figure 1). In circulation 1, a minority of pathologists confined most of the scores within a narrow range (e.g., pathologists 2, 3 and 15) compared to others (e.g., pathologists 14 and 23), and the median score for one pathologist (number 22), at 5%, was considerably lower than the median for the group. In circulation 2, the scoring across the group was more homogeneous with the majority of scores falling within a narrow range (Figure 1b). 

As observed in the heat map, the scores from two pathologists (number 7 and 20) diverged from those of others in circulation 1 but appeared somewhat similar to each other (Figure 2a). However, the pattern of scoring by pathologist 7 aligned more closely with that of the group in circulation 2 (Figure 2b) than it did in circulation 1 (Figure 2a). This change in scoring pattern by pathologist 7 between the two circulations was also reflected in a negligible agreement between this pathologist’s scores in the two circulations (Spearman ρ for absolute value = 0.156, *p* = 0.290). In circulation 2, the scoring by pathologists 4 and 13 was observed to be similar and diverged from that of others (Figure 2b). There was no zone within which inter-observer agreement was worst or best or a cutpoint, above or below which all pathologists assigned all cases in the same category, but there appeared to be fewer outlier scores above a threshold of 30% in both circulations (Figure 2). 

Examination of the scoring patterns between pathologists at FOV-level similarly showed that the scoring by a small number of pathologists (numbers 13 and 14) deviated from that of others (Appendix A). 

### 3.3. sTILs Scores in TNBC Cases

The median sTILs score for the 49 cases and the sTILs distribution between binary categories were almost identical between circulation 1 (median 23%, range of individual scores 0–97%) and circulation 2 (median 22%, range of individual scores 0–92%). The median sTILs score was <25% in over two-thirds of cases and >50% in only 10% of cases (Table 2). 

Cases with the most variation were selected on the basis of a range of sTILs scores spanning > 30% (Figure 3) and the with the highest 30% standard deviation (SD) in both circulations. This identified 21 cases between circulation 1 and 2. One pathologist (GC) reviewed the 21 cases between the two circulations to identify features that were deemed likely contributors to variation (Figure 4). The features were evaluated as described in a recent comprehensive analysis [14]. Heterogeneity of the sTILs infiltrate was defined as (a) heterogeneity at the leading edge relative to the central tumour; (b) marked heterogeneity in sTIls density within the tumour; and (c) as variably spaced-apart clusters of tumour with tight sTILs density separated by collagenous stroma with sparse infiltrate.

sTILs heterogeneity was the most common feature (11 cases) in the cases examined, followed by the presence of necrosis (9 cases), fragmentation of the cores (7 cases), difficulty delineating the boundary (5 cases), cellular tumour with low volume of stroma (4 cases), low tumour cellularity (3 cases) and technical issues e.g., out of focus, poor preservation (2 cases). In four cases, there was no obvious feature that explained the wide range of scores. When heterogeneity was present, it was mainly seen as heterogeneity in the sTIls density within the tumour (*n* = 10); or uncommonly as variably spaced apart clusters of tumour with tight sTILs density separated by collagenous stroma with sparse infiltrate (*n* = 1) or both patterns (*n* = 3). Necrosis was often focal, and zones of necrosis were generally obvious and easily avoided. Conversely, clusters of apoptosis required careful evaluation such as not to confuse lymphocytes with apoptotic tumour cells, which were the cause of the divergent score for one case (number 27) scored as 80% by one pathologist and <20% by most others. As observed by Kos et al., multiple features were often present in one case and to a variable degree (median number of features present per case, 3; range 0–4); and there was no relationship between the number of features present and variability for the case according to the standard deviation or range of scores given. Pathologists commented inconsistently on these features being present or on a difficulty with scoring those cases for which there was most variation in scores. Only a minority (median 5, range 0–12) commented on the presence of any complicating feature in these cases; and a median of only four pathologists commented for those cases with the widest range of scores (cases 24, 34, 42, 46, and 47, respectively).

sTILs heterogeneity was also present in 11 of the 28 cases for which there was less variation in scores, but it was generally minimal. A minor amount of necrosis was present in five of these cases but not to a degree that would reasonably be expected to impact scoring.

Variation in sTIls scores was examined at FOV level in the 147 FOV scored in circulation 2 to give more granular information on the source of variation in scores. There was considerable variation between the scores for the three FOV within a case, reflected by a case ICC (a measure of agreement between the three FOV from a case) that was only poor or fair in 41% (*n* = 20) and 29% (*n* = 14) of cases, respectively (Appendix A). The variability in sTILs scores across all FOV (*n* = 147) was similar to that across the cases; 29% (*n* = 43) of FOV had a >30% range of scores and a high standard deviation > 11.6 (*n* = 43) (Appendix A). 

Those FOV with a range of sTIls scores > 30% or those with 30% highest SD were reviewed by one pathologist (GC) to identify confounding factors, as was performed at case level. Again, heterogeneity within the tumour (i.e., marked heterogeneity in sTIls density within the tumour) was the most common feature (*n* = 28). Other confounding features were less common: focal necrosis (*n* = 1), scanty apoptotic cells that may have been scored inadvertently (*n* = 6), difficulty delineating the boundary (*n* = 1), low tumour stroma (*n* = 6), low tumour cellularity (*n* = 4), and technical issues (*n* = 5). Most cases exhibited one feature (median 1; range 0–3 per FOV). Thus, when other confounding factors, e.g., delineating the tumour boundary, extensive necrosis, inclusion of infiltrate around DCIS, were absent because these areas were avoided when the FOV were captured for circulation 2, heterogeneity within the tumour remained a significant contributor to variation. 

A regression model was created to provide an estimate of the proportion of inter-observer variation explained by different factors. This revealed that most of the disagreement between the sTILs scores in circulation 1 is accounted for by the variation between cases (52%) and the FOV selected by the pathologist within a case (41%) with only a minor amount due to random scoring error (7%). When these factors were controlled for in the model, agreement across the 147 FOV improved to 0.929 (compared to an ICC of 0.634 when considering only the pathologists’ average scores for each case).

## 4. Discussion

In this study, reproducibility of sTILs assessment using the internet-based scoring aid was good, but it was only fair or poor for binary categories. This was less than that reported by the TIL-WG using the same tool. However, when pathologists scored the same FOV from each case, which removed the variability associated with selecting the tumour region to score, consistency between pathologists improved to excellent. Heterogeneity of the TIL infiltrate was responsible for most of the variation, and regardless of whether pathologists selected the FOV to score or scored the same FOV, the range of TIL scores given for many cases and FOV was wide.

Early reports of reproducibility scoring sTILs emanated from studies that were designed to assess prognostic significance, mainly in TNBC and HER2 positive BC taken from clinical trials [2,3,4,5,6,7,8,9,23]. These consistently reported excellent agreement with ICC values from 0.92 to 0.97 [7,23] and 85% concordance [2], albeit between only two and three participants. Subsequent studies specifically aimed at examining reproducibility involved more participants and demonstrated good inter-observer agreement with ICC values ranging from 0.66 to 0.71 [15,16,18]. In the largest of these studies, the TIL-WG showed that good agreement was achievable between 34 participants (ICC 0.70, 95% CI 0.62–0.78) in core biopsies of TNBCs and that it improved to excellent (ICC 0.89, 95% CI 0.85–0.92) between 28 pathologists when the internet-based scoring aid was used [18]. 

To the best of our knowledge, the present study is the first, apart from the ring study, to evaluate the reproducibility of scoring sTILs using the TIL-WG internet-based scoring aid. However, agreement was less than in the TIL-WG ring study and was comparable to that reported in our previous reproducibility study without the scoring aid (ICC 0.683, 95% CI 0.601–0.767, *p* < 0.001) [15]. There are several possible explanations for this. Whilst the TIL-WG ring studies and our two studies used cores biopsies of TNBCs, case selection was different. The former used cases that had been prospectively collected from the Geparsixto trial for which sTILs data were available and adequately represented different levels of sTILs (i.e., >60% sTILs, <15% sTILs and 15–60%). Cases for the present study and our previous reproducibility study were selected to represent the spectrum of cases seen in routine practice and were consecutive cases with sufficient diagnostic material taken from one centre. It is possible that the difference in case selection contributed to the difference in the level of agreement between the studies. Second, the level of expertise of participants in sTILs scoring may have been a factor, although participants in the present study were experienced breast pathologists who had already completed a prior sTILs reproducibility study and included three members of the TIL-WG who had participated in its ring studies. When participants used the scoring aid to assess the same FOV from each case in the present study, inter-observer agreement was excellent (ICC 0.798, 95% CI 0.727–0.864, *p* < 0.001) and the lower 95% confidence level was above the minimum pre-specified value of 0.70 in the TIL-WG ring study. This suggests that subjectivity associated with selecting tumour areas or FOV to score is the main cause of variability and is supported by a regression model that showed that most of the disagreement is due to variation between cases and FOV (52% and 41% respectively). This source of variability is not mitigated using the tool and may explain why agreement using the tool to score digital whole slide images was similar to that achieved without the tool in our previous study [15] 

Agreement for sTILs across a range of cutpoints was only poor to fair and was best at 25% (ICC 0.573 95% CI 0.475–0.684, *p* < 0.001), although the series included few cases in the high-sTIL binary categories >50%. While consistency around cutpoints is generally lower than for continuous measurements [15,16,18], the TIL-WG reported substantial agreement for a range of cutpoints using their scoring aid [14,18]. In practice, the optimum threshold will depend on the clinical endpoint being examined. A cutpoint of 30% sTILs emerged as prognostic in the adjuvant setting. In a pooled analysis of 2148 early stage TNBCs treated with anthracycline-based chemotherapy, with or without taxane, there was a substantial difference in outcome for sTILs ≥ 30% across all groups and the 3-year invasive disease-free survival (DFS), distant DFS and OS for node-negative patients were 92%, 97%, and 99% respectively [5]. In a retrospective analysis of 476 patients, stage I TNBC with sTILs > 30% had an excellent outcome in the absence of chemotherapy [24]. There were fewer outlier scores at levels above 30%, and others similarly showed fewer outliers at high median sTIL scores [16,18]. Increasing 10% increments of sTILs are also associated with improved invasive DFS by 13% [5], although in our previous analysis, agreement was poorest for sTILs in 10% increments [15]. In the neoadjuvant setting, a pooled analysis of German trials demonstrated that baseline sTILs > 60% in treatment naïve samples was significantly associated with improved likelihood of pathological complete response [9]. As a potential predictive marker, an exploratory analysis of the phase II KEYNOTE-086 trial suggests that sTILs > 5% may predict response to pembrolizumab in the metastatic setting [10]. 

Heterogeneity of sTILs was the main contributor to variation in scores. Of the three patterns described of heterogeneity within the tumour was the most common and was present at both case and FOV level. Heterogeneity complicates assessment in both core biopsies and excision specimens because the former generally contains multiple separate cores and fragments of tumour. Additionally, the interpretation of sTILs hotspots poses a particular challenge [25]. Current guidance recommends averaging sTILs over the tumour and not focusing on hotspots [12]; however, this is difficult to achieve on core biopsies and the inclusion or exclusion of the hotspot impacts on consistency. In this study, the key elements of the TIL-WG scoring tool, namely, to scoring and averaging multiple areas and matching against reference images, did not mitigate the effect of heterogeneity of reproducibility and it may be that quantification of heterogeneous infiltrates proves to be a limiting factor for visual assessment of sTILs. In this regard, computational approaches are currently being explored as a more efficient and reliable methodology for sTILs assessment, which has been facilitated by the success of machine learning algorithms in pathology and the availability of large publicly available datasets for training [26]. An added advantage of computational methods is the potential to additionally address the spatial distribution of TILs and thus provide insights into the biologic and clinical significance of distribution patterns and TILs-tumour interactions. The significance of spatial distribution and hot spots has recently been examined in BC using artificial intelligence algorithms [27,28,29] and data suggest that spatial distribution may provide added prognostic information [27].

In the meantime, rigorous training is likely to improve the consistency of visual assessment of sTILs, as has been demonstrated for Ki67 in BC [30,31]. In this study, pathologists’ interpretation of sTILs guidance varied, as evidenced by the divergent scoring patterns of pathologists compared to the group within and between the two circulations; this was also observed in other studies (observer P2 [16] and P30 [18]). The lack of comments from pathologists for those cases with most variation also suggests that challenging cases may not be readily appreciated. Training in sTILs assessment that focuses on challenging cases and an emphasis on using visual calibrated references for scoring is likely to improve reproducibility. Refinements to the TIL-WG internet-based scoring aid, for example to include reference images of each 5% sTILs increment and pitfalls at the same specified magnification, will enhance its value as an educational and training resource. The time taken to select, capture, and upload FOV from each case, at 5 min, will be more acceptable for a training than a practice setting. However, once images are uploaded, the time required to score each case against the reference images, also 5 min, was similar to the time taken to score a case without the tool [15]. At present, the stand-alone graphic of sTILs reference images [12] is more easily used in a busy routine practice. 

The degree to which these issues relating to reproducibility of sTILs assessment impacts clinical decision making is unclear. This can vary for biomarkers and depends on whether absolute values or individual bins defined by cutpoint thresholds are used and the particular endpoint being examined. For example, BC tumour grading is a widely used prognosticator and has only moderate concordance between pathologists (Kappa = 0.48) [32], whereas excellent agreement (ICC 0.87, 95% CI 0.799–0.93) has been achieved for Ki67, albeit with considerable effort, but its use in BC is limited [31]. An integrated survival prediction model for early stage TNBC (tilsinbreastcancer.org) suggests that variation in absolute sTILs scores will have only a modest effect on prognostication [5]. A difference of 30% will result in less than 10% difference in predicted invasive disease-free survival in a 45-year-old patient with node-negative, grade 3, TNBC changing from 74% (95% CI 0.74–0.78) at 20% sTILs to 82% (95% CI 0.79–0.86) at 50% sTILs. However, a more stringent level of consistency will be required for sTILs as a predictive marker. Several studies have shown potential for sTILs to predict response to NACT [33,34,35] independent of chemotherapeutic agents used [36]. Furthermore, recent trial data suggest that pre- and on-treatment sTILs can have a role in the selection of patients for immune check point blocking therapies [37]. The number of lymphocytes seems to increase after only one dose of immune therapy and is associated with significant clonotype changes in T cells suggesting activation of T cells based on the expression of immune-check point, effector, and cytotoxic markers [38].

There are limitations to this study. First, fewer participants completed the second circulation than the first and thus it is possible that the observed improvement in agreement in circulation 2 was due to fewer participants. However, analysis of the inter-observer agreement between the subgroup of 14 pathologists in circulation 1 was no different to that for the original group of 23, and the second circulation also included one of the two participants who diverged significantly from colleagues in circulation 1; thus, this subgroup can be considered to be representative of the original larger group. Second, this study included a small proportion of cases with >50% sTILs, which limits the conclusions that can be drawn on the utility of the binary cutpoints when compared to the TIL-WG ring study that was designed to include equal representation of the different sTILs categories [18]. However, the cohort comprised consecutive cases and the distribution of sTILs, including the proportion of sTILs rich cases, which was representative of that in routine practice. Finally, participating pathologists did not undergo a mandatory training exercise before completing the study and it is possible that such a requirement will improve the agreement in practice. 

## 5. Conclusions

In conclusion, there is strong evidence to support sTILs as a biomarker in TNBC. Reproducibility is good for visual quantification; however, significant issues remain that limit expansion of the use of sTILs in clinical practice. The heterogeneous distribution of sTILs adds considerable subjectivity to assessment and makes standardisation difficult. Improvements in reproducibility can be achieved through rigorous training of pathologists, which can potentially be enhanced by the use of the TIL-WG internet scoring aid, and the ongoing efforts of the WG in standardising sTILs assessment. The process followed by the International KI67 in Breast Cancer WG to standardise Ki67 can be used as a template in this endeavour [31]. However, ultimately, computational and machine-learning based approaches may be needed to achieve an acceptable level of analytic validity if the clinical potential of sTILs as a biomarker is to be fully realised.

## Figures and Tables

**Figure 1 cancers-13-04410-f001:**
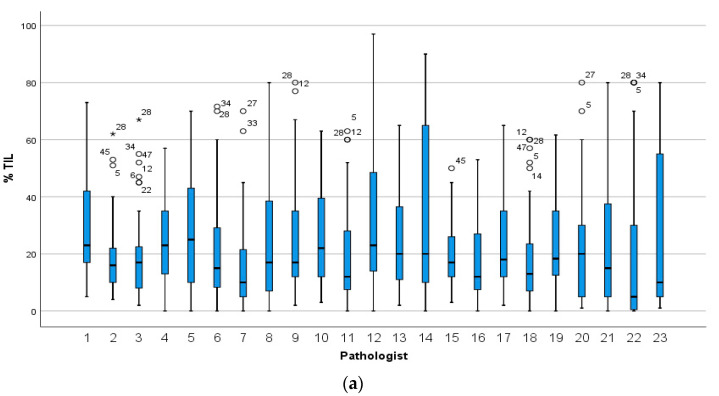
The distribution of sTILs scores for the 49 cases given by (**a**) 23 pathologists in circulation 1 and (**b**) the 14 pathologists in circulation 2. Pathologists are numbered on the x-axis and those who completed both circulations are assigned the same number in circulation 1 and 2. Stars and circles are outliers which lie abnormally far from the other values in the dataset. A circle denotes a value which is either greater than 3rd quartile plus 1.5× interquartile range or is less than 1st quartile minus 1.5× interquartile range. A star denotes an extreme outlier value, which is either greater than 3rd quartile plus 3× interquartile range or is less than 1st quartile minus 3× interquartile range. The number beside the stars and circles is the TNBC slide and case number for that outlier. There was more variation in scoring when each pathologist selected the tumour area to score (**a**) compared to when each pathologist scored the same FOV (**b**).

**Figure 2 cancers-13-04410-f002:**
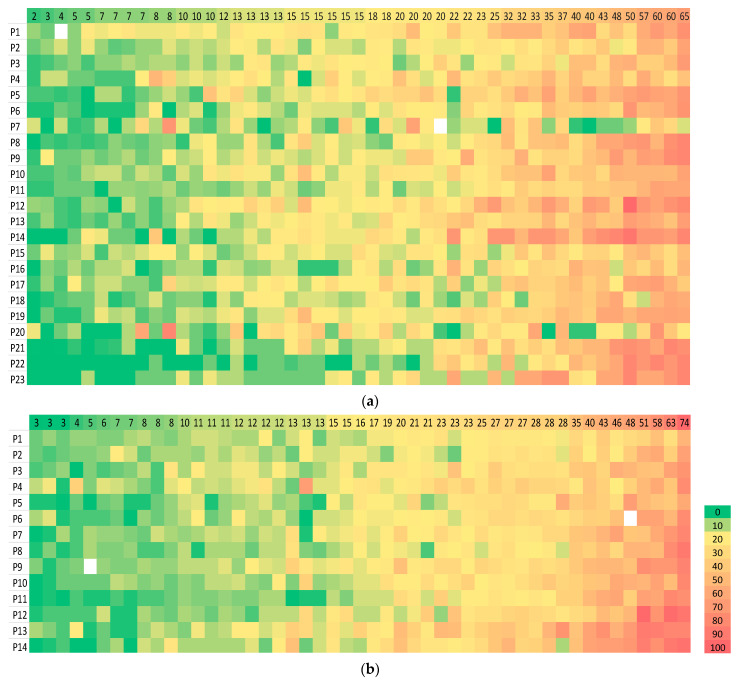
Heatmap of sTILs scores given by pathologists for 49 TNBC cases. A visual representation of sTILs scores in which each row represents a pathologist (numbered as in Figure 1) and each column a case for (**a**) Circulation 1 and (**b**) Circulation 2. The colour of each cell represents the sTILs score for the case, increasing from low (green) to high (orange), as denoted in the separate vertical key; missing data are white. The value given in each cell in the top row of each heat map is the median sTILs score for that case in the respective circulation, and the cases are ordered left to right from the lowest to the highest median sTILs score. Outlier scores for a case relative to others appear as mismatches of colour e.g., green cells within a zone of orange. There is more heterogeneity of scores in circulation 1 compared to circulation 2 and the scores given by pathologists 7 and 20 in circulation 1 diverge considerably from others, whereas the scores given by pathologist 7 in circulation 2 align with those of other pathologists. In circulation 2, the scoring pattern of pathologists 4 and 13 appear to diverge from that of others.

**Figure 3 cancers-13-04410-f003:**
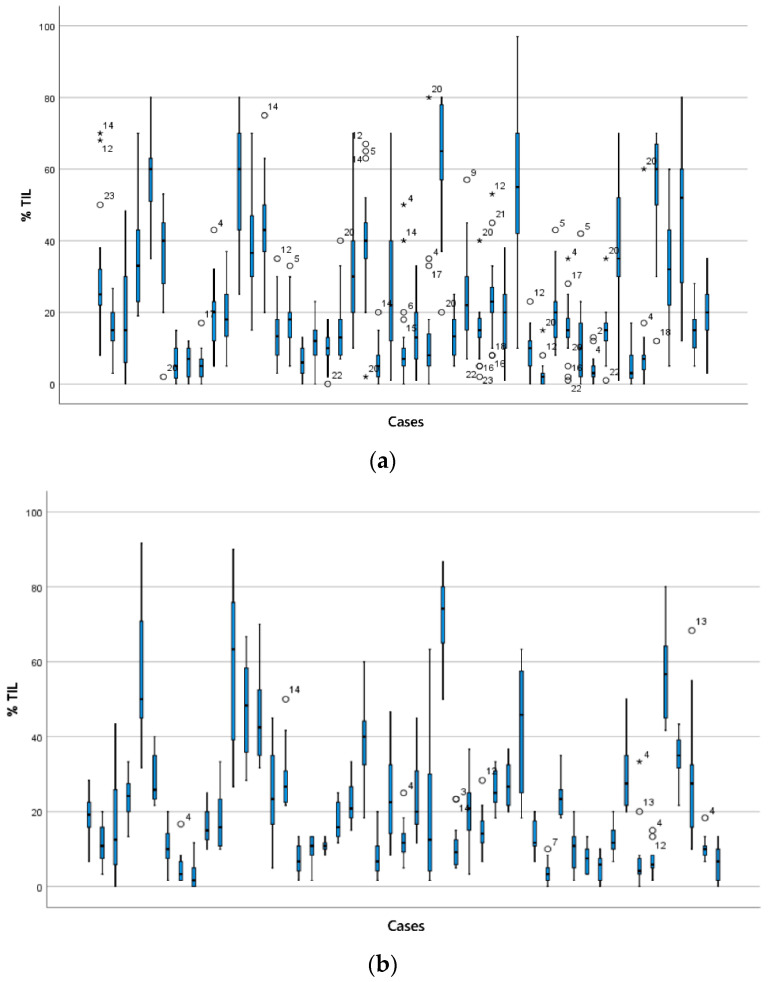
sTILs scores for TNBCs. The range of sTILs scores is shown for the 49 cases in (**a**) circulation 1 and (**b**) circulation 2. Cases are displayed from case 1 (left) to 49 (right) on the x-axis in (**a**,**b**). Stars and circles are outlier values which lie abnormally far from the other values in the dataset (as defined for Figure 1). The number beside each star and circle is the number of the pathologist for that outlier. The range of scores is wide for many cases.

**Figure 4 cancers-13-04410-f004:**
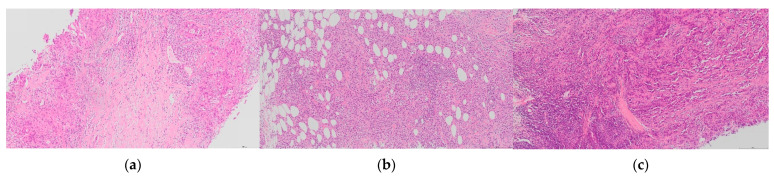
H&E images of sTILs heterogeneity in TNBC. Sparse sTILs within stroma between tumour with more dense sTILs ((**a**), case 21); dense aggregates of sTILs within tumour ((**b**), case 24); and variation of sTIls across a tumour with an indistinct tumour boundary in which sTILs are most dense ((**c**), case 42). A wide range of sTILs scores was given for these cases (65× magnification).

**Table 1 cancers-13-04410-t001:** Inter-observer agreement between pathologists scoring sTILs in circulation 1 and 2.

sTILs	Circulation 1 ^a^ (*n* = 23)			Circulation 1 ^a^ (*n* = 14)			Circulation 2 (*n* = 14)		
(%)	ICC	95% CI	*p* Value	ICC	95% CI	*p* Value	ICC	95% CI	*p* Value ^b^
Absolute values									
	0.634	0.539–0.735	<0.001	0.662	0.566–0.760	<0.001	0.798	0.727–0.864	<0.001
Cutpoints									
≥10 ≥20	0.368 0.484	0.278–0.488 0.385–0.603	<0.001 <0.001	0.426 0.492	0.324–0.551 0.388–0.614	<0.001 <0.001	0.486 0.657	0.382–0.608 0.561–0.756	<0.001 <0.001
≥25	0.573	0.475–0.684	<0.001	0.602	0.501–0.711	<0.001	0.596	0.495–0.706	<0.001
≥30	0.537	0.438–0.652	<0.001	0.555	0.451–0.671	<0.001	0.597	0.495–0.706	<0.001
≥40	0.488	0.389–0.606	<0.001	0.506	0.401–0.626	<0.001	0.644	0.546–0.745	<0.001
≥50	0.428	0.332–0.549	<0.001	0.440	0.337–0.564	<0.001	0.569	0.465–0.682	<0.001
≥60	0.351	0.263–0.470	<0.001	0.368	0.271–0.493	<0.001	0.479	0.375–0.602	<0.001

^a^ For circulation 1, inter-observer agreement is given for all 23 pathologists and separately for the 14 pathologists who undertook circulation 2. ^b^ A statistically significant value for the absolute values and cutpoints supports the interpretation that there is sufficient evidence in the sample to conclude that the relationships exist in the population. CI, confidence interval; ICC, intra-class correlation coefficient.

**Table 2 cancers-13-04410-t002:** Distribution of TNBC Cases According to Binary Categories Defined by sTILs Cutpoints.

sTILs	*n* (%) ^a^	Cutpoint	*n* (%) ^a^
Cutpoint			
<20%	27 (55)	≥20%	22 (45)
<25%	34 (69)	≥25%	15 (31)
<30%	35 (71)	≥30%	14 (29)
<40%	40 (82)	≥40%	9 (18)
<50%	44 (90)	≥50%	5 (10)
<60%	46 (94)	≥60%	3 (6)

^a^ The number of cases in each of the binary categories is given according to different sTILs cutpoints.

## Data Availability

The data presented in this study are available in the article and in Appendix A.

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
