# Peer review of "Intra-Tumour Heterogeneity Is One of the Main Sources of Inter-Observer Variation in Scoring Stromal Tumour Infiltrating Lymphocytes in Triple Negative Breast Cancer"

_cancers, 2021, doi:10.3390/cancers13174410_

Round 1

Reviewer 1 Report

I thank the authors for the time and effort spent exploring my suggestions. I think some of the analyses were fruitful and have added to the originality and impact of the article. Great job on this important work. Some minor typographical recommendations remain-

  • Sentence 2 abstract please revise grammar
  • Spelling “focussing” p3 151

Reviewer 2 Report

With all the corrections made I think this paper is ready for publication.

Reviewer 3 Report

Reviewer has no further questions.

This manuscript is a resubmission of an earlier submission. The following is a list of the peer review reports and author responses from that submission.

Round 1

Reviewer 1 Report

The authors present a study were an internet-based scoring aid was used to assist with evaluation of stromal tumor infiltrating lymphocytes (sTILs). Twenty-three pathologists across Europe evaluated sTILs in 49 TNBC consecutive cases. The study showed that intra-tumor heterogeneity is a source of inter-observer variation in scoring which make standardization difficult. The authors propose training of pathologists for standardization of sTILs assessment, which may potentially be improved using automated approaches.

The paper is clear and well written and easy to follow. The conclusions are consistent with the arguments presented.

A couple of minor comments:

I would like the authors to present the patient material in a “classical” table 1, with the distribution stated in percentage for all variables, and not just for some of them which is done now in section 2.1.

The is a typing error on line 152, “reference images” is repeated twice, please correct.

In Table 1 the is an a hyphenated, but I cannot see the explanation for this. Please correct.

In line 230-231, the authors claim that the pattern of scoring for pathologist 7 aligned with that of the group in circulation 2, is that correct? How can you see that? I would say that pathologist 7 do NOT align, especially not in circulation 1. This sentence needs to be rephrased.

For Table 2 I would assume that the binary categories are connected two and two? If so, it would be better if they were presented side-by-side instead of underneath each other.

The sentence underneath Table 2, is that a description connected to the table? A footnote seems to be missing.

Reviewer 2 Report

This manuscript is clear and well written. A few minor comments:

  1. Quantitative reproducibility scores/metrics were often described using qualitative attributes (e.g., excellent; good; fair; poor). These qualitative attributes seemed to be based on numerical cutoffs. Please define those numerical cutoffs in the manuscript.
  2. In several places, the authors mention “potential of tool for wider application”. It would be helpful if the anticipated application is described immediately after the statement.
  3. The number of years (e.g., median and range) of breast pathology experience for the pathologist should be noted.
  4. The term “images” was used to describe the entities assessed for Circulation 1 and Circulation 2. This is confusing because the generic term “images” does not appropriately capture the nuances in the entities assessed in Circulation 2. Specifically, whole slide images of the full biopsies were scored for Circulation 1. However, Fields of View (FOV) or Regions of Interest (ROI) are the entities that are scored for Circulation 2. This terminology (i.e., FOV or ROI) should be used throughout the manuscript when describing images that were assessed for Circulation 2. This helps the readers to better understand the distinctions between Circulation 1 and Circulation 2 and therefore better understand the purpose of the study and impact of the results.
  5. Page 3 line 19, please add literature citation for the study mentioned.
  6. Table 1. All values were statistically significant (p<0.001) for the absolute values and each of the cutpoints. What is the authors’ interpretation of such results?
  7. Figure 1 legend. Line 246 has an extra parenthesis in the line.
  8. Figure 3. Please label the x-axis.
  9. In the discussion, the authors mention that the pathologist agreement in this study was less than that of the ring study but was comparable to their previous reproducibility study without the scoring aid. The comparison with the previous study suggests that the scoring aid was not helpful. Could the authors please elaborate on that finding.

Reviewer 3 Report

Strengths: This is a useful and straightforward assessment of TILs concordance. Two experiments are conducted, first an experiment assessing sTIL concordance across 49 cases and 23 pathologists using methods and tool provided by the TIL-WG. The second is an experiment repeating the assessment using pre-selected H&E images (n=3) for each sample. Overall ICC are reported for experiment 1 and experiment 2, as well as means and distributions of sTIL scores, by pathologist, and by sample. Also, ICCs are calculated for various binary sTIL cutoffs. This experiment adds additional perspective on TILs concordance, to the previously published data from the RING studies and other studies.

Weaknesses and things that need to be addressed:

  • Complete details of the methods for calculating ICC, and rationale for selection these methods, are not provided in the manuscript. Also, there are different types of ICCs that could be explored (i.e. versions that assess intra-rater consistency versus absolute agreement), to make the manuscript more substantial, and to illustrate how much discordance is related to systemic error of certain pathologists
  • Provide more detail on how the 3 standardized images were selected for circulation 2. Were “low TIL” “med TIL” “high TIL” zones sampled? Was it intended to be random? Is there good correlation in overall TIL score for circulation 1 v. 2?
  • For circulation 2, additional, more nuanced analyses could be pursued by evaluating more granular data pertaining to sTIL estimates across each H&E image, rather than focusing exclusively on the mean/average sTIL score across the 3+ images per sample. Examples of potential analyses include:
    • Calculating ICCs for each of the 3 pre-selected H&E images, followed by scrutinizing the H&E images that had lower than expected ICC
    • Evaluate whether image-level ICCs vary according to factors such as location of the image (i.e. near invasive edge of tumor versus center of tumor), inclusion of necrosis or apoptosis, proportion of stroma
    • Create a formal (and less subjective) definition of intratumoral heterogeneity, and explore the impact of this definition on overall sample-level ICC. For example, you can define intratumoral heterogeneity by the coefficient of variation of mean sTIL score estimate across each of the 3 images in the given sample.
    • You could calculate the deviation of an individual pathologist’s score versus the mean score, for individual H&E images, and see if you could compute deviation scores for each of the pathologists to see if there are any outliers in this regard,
  • The title of the manuscript is quite assertive, implying a quantitative analysis of the determinants of intra-observer variance. However, in fact, the analysis was subjective, relying primarily upon a relatively vaguely-described subjective classification of features that were deemed likely contributors of variation (page 8, 274). Before publication, more detail must be provided in this section so that the experimental methods could be reproduced.
    • I.e. did multiple pathologists classify these features? What are the strict definitions of these features? how is “heterogeneity of the sTILs infiltrate” defined? what happened in cases with more than one feature, or when features are present in some but not all of the H&E images for the sample?
    • Also the title should be made less assertive, so the paper delivers on its promise
    • It might be possible to collaborate with a statistician to create a regression model or another statistical simulation methods that can more elegantly and quantitatively estimate the proportion of inter-observer variance that can be explained by various factors, including 1) intra-observer systemic error; 2) intra-observer random error; 3) image selection (variation in images selected, i.e. circulation 1 v. 2), 4) quantity of images selected (i.e. comparing ICC if image numbers are downsampled or upsampled).
  • Sample size is quite small, and the confidence intervals for ICCs estimations across different binary endpoints are high, and this should be noted in the discussion section when commenting on the utility of various binary cutpoints, and limitations of the experiment
  • Page 11 374- unfair to cite KEYNOTE 086 in this section because its METATSTATIC TILs, which is different than early stage TILs
  • Page 11 380. It is inaccurate to say that hotspots are excluded from analysis. In the WG guidelines, they recommend including hotspots in the overall average, but not focusing on them.
  • In the discussion section, it should also be noted that the sTIL score in some settings is used as a pharmacodynamic biomarker of treatment effect, for example some clinical trials aim to show an increase in TIL score associated with therapy. If so, interobserver concordance would be vital for success in this type of endeavor
